# Gate controlled valley polarizer in bilayer graphene

Hao Chen[1,2], Pinjia Zhou[1], Jiawei Liu[1,2], Jiabin Qiao[1], Barbaros Oezyilmaz[1,2] & Jens Martin ![ORCID] [1,2,3 ✉]

Sign reversal of Berry curvature across two oppositely gated regions in bilayer graphene can give rise to counter-propagating 1D channels with opposite valley indices. Considering spin and sub-lattice degeneracy, there are four quantized conduction channels in each direction. Previous experimental work on gate-controlled valley polarizer achieved good contrast only in the presence of an external magnetic field. Yet, with increasing magnetic field the ungated regions of bilayer graphene will transit into the quantum Hall regime, limiting the applications of valley-polarized electrons. Here we present improved performance of a gate-controlled valley polarizer through optimized device geometry and stacking method. Electrical measurements show up to two orders of magnitude difference in conductance between the valley-polarized state and gapped states. The valley-polarized state displays conductance of nearly $4e^2/h$ and produces contrast in a subsequent valley analyzer configuration. These results pave the way to further experiments on valley-polarized electrons in zero magnetic field.

[1] Centre for Advanced 2D Materials, National University of Singapore, 6 Science Drive 2, 117546 Singapore, Singapore. [2] Department of Physics, National University of Singapore, 2 Science Drive 3, 117551 Singapore, Singapore. [3] Present address: Leibniz Institut für Kristallzüchtung, Max-Born-Strasse 2, 12489 Berlin, Germany. ✉email: jens.martin@ikz-berlin.de

The valley degree of freedom in graphene originates from the hexagonal structure and may be exploited in electronic devices. Besides the formation of beam splitters[1,2] and waveguides[3,4], valley filters[5] are in particular interesting, as they enable experiments to investigate electronic transport in the presence of non-zero Berry curvature[6]. In general, valley currents are not easily detected. Usually, a non-local geometry is adopted and the inverse valley Hall effect is exploited[7,8] to measure the valley Hall effect[6,9]. Instead, valley-polarized electrons would allow a direct measurement of the valley Hall effect. Further, one can imagine experiments in which the sub-lattice pseudo-spin is manipulated and detected solely by electrical control[10].

Experimentally, valley currents can be created by circularly polarized light[11,12], magnetic control[13,14], or as counter-propagating bulk current[7,8,15]. However, optical and magnetic control is not always feasible and net bulk valley currents offer little flexibility in control. Instead, gate-controlled chiral channels could provide a robust and flexible approach for valleytronic operations. Theory predicted that[10,16–18] sign reversal of Berry curvature in adjacent regions in bilayer graphene (BLG) offers a scheme to produce topological valley-polarized one-dimensional (1D) channels. This can be achieved by dual-split gates (Fig. 1a). Each gate pair operates independently and induces a gapped state within the gated region of the BLG. At the interface of the left and right gate pair, an artificial grain boundary is formed for odd-field configurations, defined such that for the displacement fields $D_R$ and $D_L$: $D_R \cdot D_L < 0$. In this case, along the split junction there exist valley-polarized 1D channels with quantized conductance $4e^2/h$ in the ballistic limit (Fig. 1b). Each direction is valley-polarized and consists of fourfold conduction channels due to spin and sub-lattice degrees of freedom. In the evenly gated case, where $D_R \cdot D_L > 0$, the junction will be in a regular insulating gapped state.

Previous work at naturally occurring stacking boundaries demonstrated good contrast between the different electric field configurations[19,20]. However, the first experiment on a gate-controlled polarizer was far from ballistic transport[21]. Instead, the contrast between oddly and evenly gated configurations was limited due to intrinsic disorder causing valley mixing and backscattering. This renders the 1D channels significantly more resistive. Good contrast between even and odd-field configuration was achieved only in the presence of an external magnetic field[1]. The external magnetic field helps[18] to spatially separate the counter-propagating channels and hence suppress backscattering. At high magnetic field of 8 T, the work demonstrated nearly ballistic transport confirmed by the observation of four quantized conductance channels in the odd-field configuration. Yet, the application of high magnetic fields may limit the use of valley-polarized electrons for further studies on electronic transport in the presence of non-zero Berry curvature.

Here we demonstrate improved performance of a purely gate-controlled valley polarizer at zero magnetic field employing optimized geometric dimensions and sample stacking schemes. In valley polarizer regime, conductance between valley-polarized state and gapped state shows two orders of magnitude difference. Evidence of chiral nature of the 1D channels is provided by measurements in valley analyzer configuration.

## Results

**Electrostatic simulation of device geometry**. We start with a discussion on optimizing device geometry. We consider the effect of device asymmetries in real devices and their effect on introducing charge inhomogeneity within the channel. For simplicity, we limit the discussion to the case of even displacement field configuration.

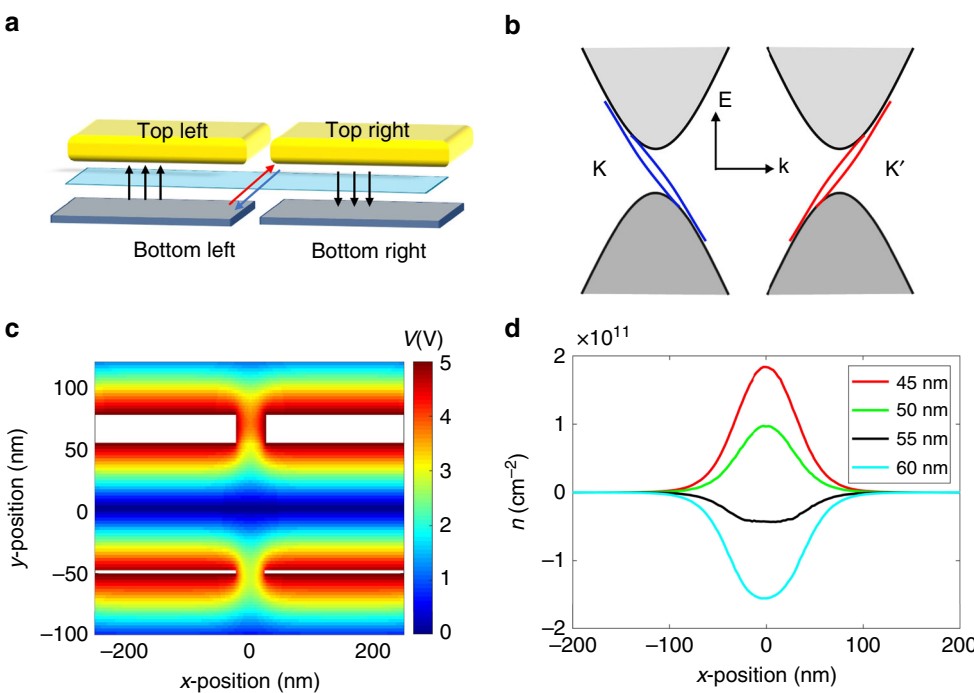

**Fig. 1 Schematics of device geometry and electrostatic modeling. a** Schematic of the dual-split gated device. Two pair of gates tune independently the displacement field and charge density in the left and right part. **b** Schematic of valley-polarized channels K and K' in each Dirac cones, with opposite group velocity. **c** Cross-section view of electrical potential from electrostatic modeling. The horizontal axis corresponds to x direction position and vertical axis corresponds to z direction position. **d** Distribution of residual charges density n in bilayer graphene in an asymmetric device. Different spacing parameters of top gates are tested to minimize the residual charge density when bottom gates spacing is constant 50 nm.

The first imperfection with respect to gating homogeneity is due to the presence of a non-zero gap between the gate pairs. In this region the gating efficiency is reduced and hence the electric field-induced gap is reduced (Fig. 1a). Due to thermal excitation and intrinsic disorder potential, this is expected to reduce the resistance of the insulating state. Further, the lateral confinement can produce additional sub-bands[22–24], which are partially located inside the bandgap, reducing the effective gap even further. Yet, as long as disorder is negligible, sufficiently low temperature will produce insulating behavior. In contrast, in real devices there are many causes for disorder.

We are particularly interested to estimate the amount of charge disorder created by asymmetric gating. Asymmetry with respect to the upper and lower half, as well as the right and left half of the dual-split gate, can be caused by various means. Clearly, misalignment of top gates with respect to bottom gates is problematic. In practice, however, alignment accuracy of much better than 5 nm can be achieved in electron beam lithography (EBL), which limits this effect to an affordable level (see details in Supplementary Information). What is more difficult to handle is achievement of equal thickness of the top- and bottom-gate dielectric. In manually stacked samples, hexagonal boron nitride (hBN) flakes of similar thickness are selected, but it is very challenging to find hBN flakes with exactly the same thickness. Further, we choose atomically thin graphene bottom gates to maintain the lower hBN layer as flat as possible. For the top gates, we remain with evaporated Au gates to limit the number of two-dimensional transfers.

To quantify asymmetric gating efficiency, we consider a simple electrostatic model using finite element methods (see Supplementary Figs. 1–5 for more details). Gates are assumed as metals with rectangular cross-sections. We use 2.5 nm for the few-layer graphene bottom gates and 25 nm for the metal top gates. BLG is also assumed to be a metal, which will yield an upper bound of the induced charge. By varying the device geometry, we calculate the induced charge on BLG in the split region for the case that the charge neutrality is maintained in regions directly underneath the gates. Further details can be found in Supplementary Information.

As a guideline, the gate-induced charge disorder should remain smaller than intrinsic disorder caused by charge impurities or strain. In typical encapsulated graphene devices, the intrinsic disorder amounts to $\Delta n_{disorder}$ in the range of $10^{10}$ cm$^{-2}$ [7,25]. We find that asymmetric gating can easily exceed this, but gate-induced charge disorder can be minimized by adjusting the top-gate geometry according to measured device parameters. In practice, bottom-gate spacing and thickness of the dielectrics can be experimentally determined on the real device before fabricating the top gates. This means that a careful adjustment of the top-gate spacing may really improve device performance.

To illustrate the optimization procedure, we discuss the hypothetical case where the graphene bottom-gate spacing is designed to be 50 nm, as well as the thickness of the dielectric hBN layers between the gates and BLG. The top gate is designed with 25 nm thickness causing the device to be asymmetric, as the bottom gates are much thinner. Then, we pose the question what top-gate spacing minimizes the induced charges in the split region? In Fig. 1d, we show the effect of varying the width of the gap in the bottom gates.

Calculating the induced charges with the same 50 nm spacing in top gates results in a maximal excess of 3% charge carriers with respect to the charges induced by the bottom gate. At typical gate voltage of 5 volts, this would correspond to a charge density of $10^{11}$ cm$^{-2}$, which is larger than the typical charge-carrier fluctuations in hBN-encapsulated devices. By carefully adjusting the spacing, the induced charge carriers can be reduced by a factor of 5, which is a significant amount and bring the level of gate-induced disorder into a range that is comparable to the expected charge-carrier fluctuations already present in the device.

Although this demonstrates that careful adjustments to the device design, based on specific measurements of the device geometry during the fabrication process, can improve device fabrication, the results also show that gate geometry must be precisely defined in the nanometer range. This remains a challenging task considering typical edge roughness in the nanometer range, possible misalignment, and fabrication-borne effects such as proximity and overexposure.

**Device fabrication**. The actual device is fabricated by stacking hBN, BLG, and hBN layer subsequently and release this stack atop pre-patterned few-layer graphene split gates. Samples are designed to have four gate pairs to increase yield and to allow for valley analyzer experiments. A dry van der Waals stacking method is exploited to reduce inhomogeneity and disorder in the heterostructure. In general, the split channel width of around 50 nm is fabricated. The graphene bottom gates are etched in O$_2$-plasma using a poly(methyl methacrylate) (PMMA) mask. We have optimized the fabrication to accommodate dimensional changes in the real device with respect to the design. For example, etching of the graphene bottom gates affect the width of the gates depending on the etch time, concentration, etc. Further, we exploit a slight overdose for the top gates to minimize PMMA residues, but increases the gate width of 15–20 nm. Hence, to achieve a nominal gate spacing of 50 nm, we use a 35 nm gap for the bottom layer and a 65 nm gap for the top layer. Edge contacts[26] are used to contact the fully encapsulated BLG. The design of the top gates is adjusted as discussed above. Further details of device fabrication are included in Methods and Supplementary Information (see Supplementary Table 1 and Supplementary Figs. 6 and 7 for more details).

**Electrical measurement**. The electrical measurement setup is illustrated in Fig. 2a (details in Supplementary Information). Figure 2b shows an optical image of the complete device. Figure 2c is a scanning electron microscopy (SEM) image of the bottom gates. The split gaps are well fabricated and controlled with 50 ± 5 nm. Electrical measurements are performed at $T \sim$ 1.4 K unless otherwise stated. Field-effect mobility of around 150,000 and 320,000 cm$^2$ V$^{-1}$ s$^{-1}$ is achieved in device 1 and device 2, respectively, revealing high sample quality (see Supplementary Fig. 8 for more details). The mean free path of electron is estimated to be around 330 nm and 700 nm for device 1 and 2, respectively. This is comparable to gate length (400 nm), indicating ballistic transport across the gates.

Figure 3 demonstrates for device 1 the formation of chiral 1D states in the odd gating condition. To demonstrate most convincingly the switching behavior between insulating and ballistic 1D channels, we fix the voltages of one half of the split gates. We choose the voltages such that the resistance in the gated region is maximized (see Supplementary Information). Then, we scan in the other two gates (right side) and map out the resistance of BLG (Fig. 3b). For convenience we plot the relevant physical scales of charge-carrier density ($n$) and displacement field. Here, the relations of $n$ and $D$-field are given by $n = \varepsilon_0 \varepsilon_r ((V_t - V_{t0})/d_t + (V_b - V_{b0}/d_b))/e$, $D = \varepsilon_r ((V_t - V_{t0})/d_t - (V_b - V_{b0}/d_b))/2$. Here, $\varepsilon_0$ is vacuum electric constant, $\varepsilon_r$ dielectric constant of hBN, $V_{t(b)}$ top(bottom)-gate voltage, $V_{t0(b0)}$ top(bottom)-gate voltage offset, $d_{t(b)}$ top (bottom) thickness of dielectric, and $e$ the elementary charge (further details are provided in Supplementary Information).

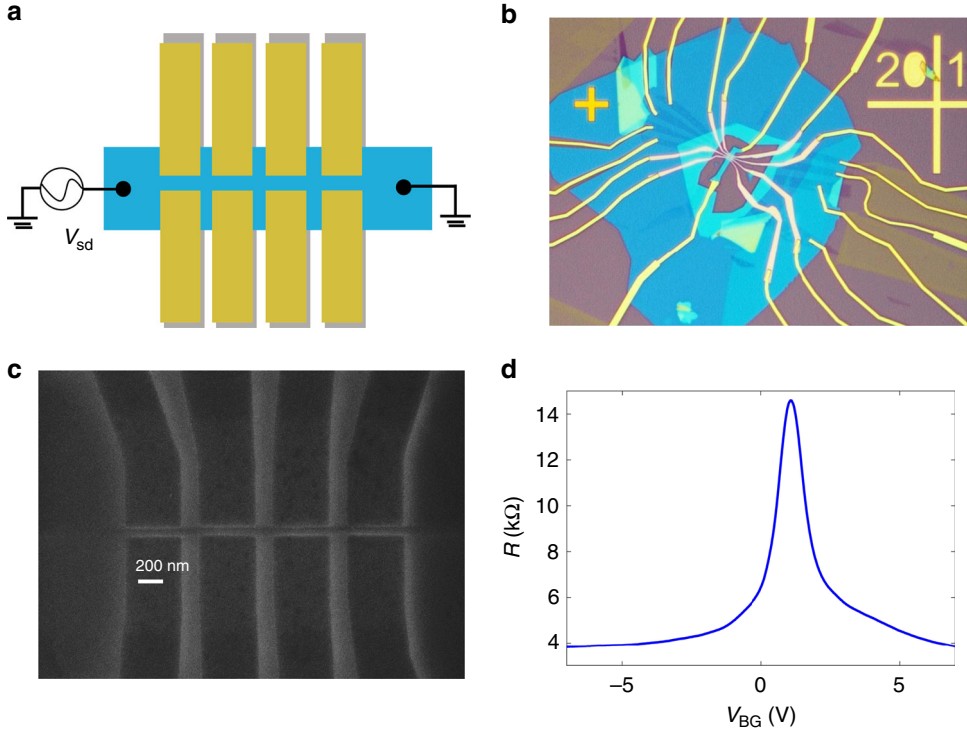

**Fig. 2 Device characterization. a** Device schematic and electrical measurement setup. **b** Optical image of complete device. **c** Exemplary scanning electron microscopy (SEM) image of bottom graphitic gates; scale bar is shown. **d** Back-gate dependence of BLG sample resistance.

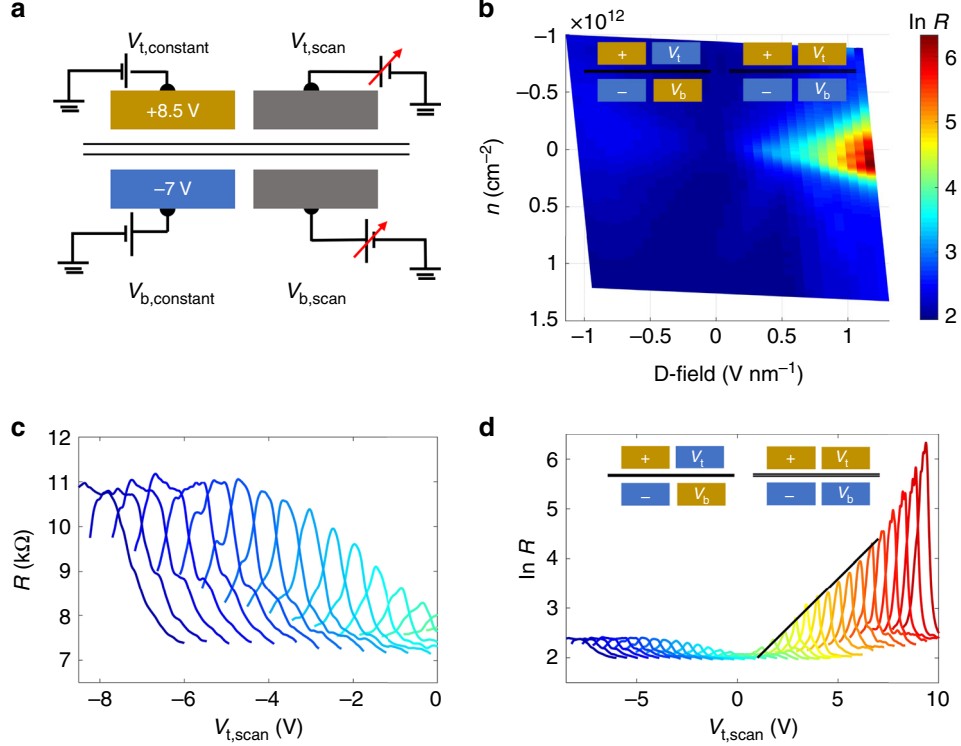

**Fig. 3 Formation of chiral states: measurement results when the left gates are fixed at a positive polarity, from device 1. a** Cross-sectional view of measurement setup, the left part is kept at constant potential, with Fermi level at CNP. Potential in the right part is varied. **b** BLG resistance (in log scale) color plot as a function of displacement field and charge-carrier density $n$. **c** Zoom-in view of the left side of **d**, corresponds to oddly gated configurations. **d** BLG resistance (in log scale) as a function of the top-gate potential when the bottom gate is fixed at different potentials. Peak values of curves grow linearly with increasing $D$-field with a straight line as a guide to the eye.

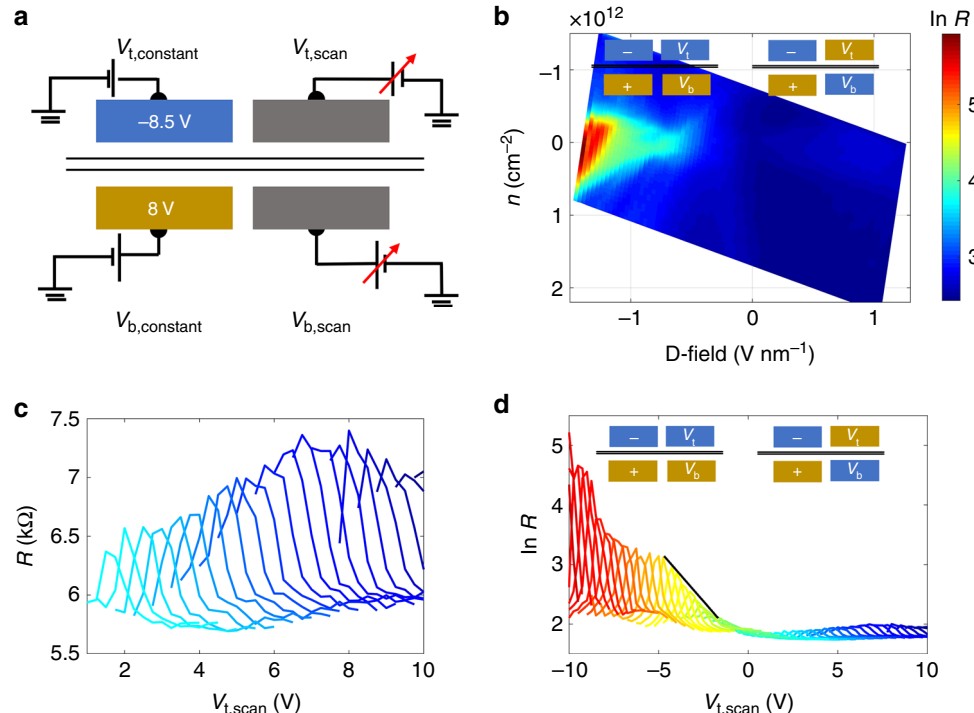

**Fig. 4 Formation of chiral states with opposite direction: measurement results when left gates are fixed at negative polarity, from device 1. a** Cross-section view of measurement setup, the left side displacement field is opposite to that in Fig. 3. **b** BLG resistance (in log scale) color plot as a function of displacement field and charge-carrier density $n$. **c** Zoom-in view of right side of **d**, corresponds to oddly gated configurations. **d** BLG resistance (in log scale) as a function of the top-gate potential when the bottom gate is fixed at different potentials. Peak values of curves grow linearly with increasing $D$-field with a straight line as a guide to the eye.

For each value of displacement field, maximal resistance is achieved in a charge-neutral point. Clearly, there is an asymmetry visible with respect to the direction of the displacement field of the right half. In the odd-field configuration, the device resistance increases slowly and then saturates at a plateau value of about 11 kOhm (Fig. 3c). In contrast, the device resistance in the even field configuration increases exponentially (linearly in log plot) reaching a maximal value of 600 kOhm at $D \sim 1.1\,\mathrm{V\,nm^{-1}}$. From the increase of the resistance, we estimate an energy gap of $\sim 0.69$ meV at $D \sim 1.1\,\mathrm{V\,nm^{-1}}$. The observed transport gap is comparable to the previously measured transport gaps on silicon dioxide substrate $(2.2\,\mathrm{meV}$ @ $2.5\,\mathrm{V\,nm^{-1}})$[27] and does not yet achieve the values of suspended BLG $(0.087\,\mathrm{V\,nm^{-1}}, \Delta = 0.31\,\mathrm{meV})$[28]. This mismatch may seem surprising, because mobility in the locally gated BLG device is much higher than BLG on $\mathrm{SiO}_x$ substrate. Possible reasons for the reduced transport gap are gating inhomogeneity as discussed above and short channel length, making the device more sensitive to intrinsic disorder.

To demonstrate the robustness of the device, we switch the polarity of the fixed gates (Fig. 4). Similar behavior is observed, except that the polarity in gate–gate scan switches due to reversed displacement field direction in the left part. Very similar as above, we observe saturation of the resistance at zero charge-carrier density in the odd-field configuration and the resistance ratio between odd and even field configuration reaches around 100. The gap for the insulating part reaches 0.42 meV at $\sim 1.4\,\mathrm{V\,nm^{-1}}$.

Data from supplementary gate configurations for device 1 (Supplementary Fig. 9) and additional device 2 (Supplementary Fig. 10) and are shown in Supplementary Information. Supplementary Table 2 shows measured $R_{\mathrm{insulating}}$ (maximum insulating resistance) and $R_{\mathrm{chiral}}$ (chiral channel resistance) from our devices and from previously published[1,21] data.

## Discussion
To estimate the channel conductance, we need to subtract the series resistance due to the ungated sheet resistance of BLG and contact resistance at the BLG/metal interface (Supplementary Figs. 11 and 12). As we are measuring in two-terminal geometry, we need to make several assumptions.

First, we assume that the resistance of the locally gated region at high density can be neglected against the resistance of the ungated BLG region and the interface resistance to the metal contacts. This seems justified, because the resistivity of highly gated BLG is approximately a factor 10 smaller than near-charge neutrality. Further, the device geometry is such that the gated region is significantly less than a square. Yet, by assuming the measured resistance at high gate voltages constitutes the correct series resistance, subtracting it from the data yields channel resistance in the odd configuration, which is slightly smaller than expected for four ballistic channels. Specifically, we derive $R_{\mathrm{high\_density}} \sim 5.48\,\mathrm{kOhm}$ (Supplementary Fig. 11). This is reasonable, but somewhat too high, considering about 5.2 squares of BLG and contact length of $1.5\,\mu\mathrm{m}$. Typical numbers for edge contacts are 1000 ohm $\mu\mathrm{m}$[26] and resistivity is about 300 ohm$^{-2}$[28] at high density yielding about 2 kOhm for above geometry. Taking the full $R_{\mathrm{high\_density}}$ as the series resistance and subtracting it from the data, we derive at a channel resistance in the odd-field configuration of 5.72 kOhm (see Supplementary Information). This is not far from the expected value $(h/4\mathrm{e}^2 \cong 6.45\,\mathrm{kOhm})$ for the 1D chiral channels, but about 15% more conductive than expected. This may indicate parallel conduction originating from inhomogeneous gating or disorder. Yet, the leakage channel seems to occur only in the odd-field configuration, as we achieve an on/off contrast of around 100, an order higher than previous reported results, making this an unlikely explanation.

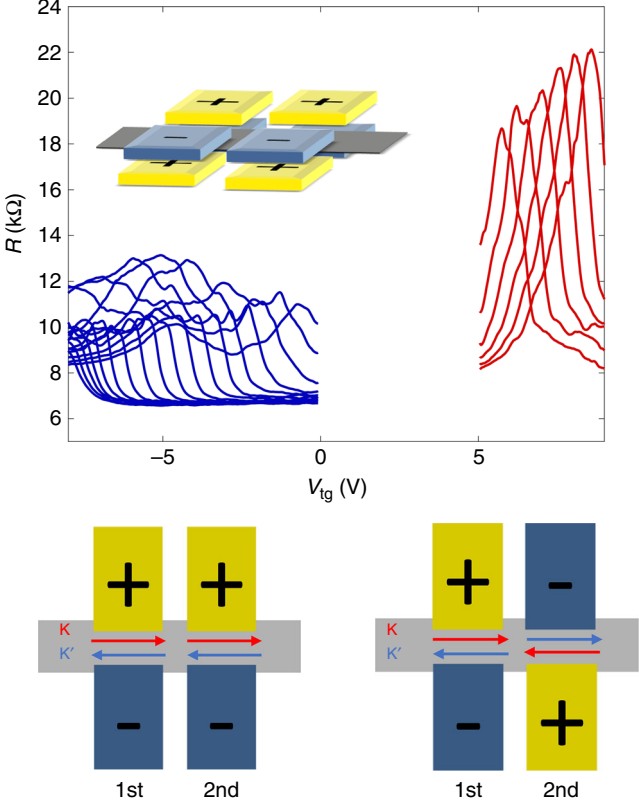

**Fig. 5 Demonstration of chiral nature: valley analyzer measurement with two gate pairs in same (left) or opposite (right) valley-polarized state.** Gate configurations are shown underneath. During the measurement, the first gate pair is kept at constant voltages to maintain in a fixed valley-polarized state. One half of the second gate pair is fixed too and only one top gate and one bottom gate is scanned (similar scheme to Figs. 3 and 4). The BLG resistance as function displays two different regimes: high (low) resistance states are observed under opposite (same) valley-polarized state of two gate pairs, indicating the chiral nature of states in odd-gate configuration. The insets above measurement curves show exemplary 3D side view of gate configurations.

At this point we note that the device resistance at high electron and high hole concentration is not the same, displaying differences up to 1 kOhm (see Supplementary Fig. 9). Indeed, we expect that an *np*-junction in BLG displays increased resistance due to anti-Klein tunneling[29]. Related to this, it has been observed experimentally[30] that *pn*-junctions display higher resistance in the *pn*-state, compared with *nn'*-state. In general, the two-terminal resistance of locally gated BLG channels depends on the combination of *p*- and *n*-regions, and the size of locally induced gaps[31,32].

To derive the correct series resistance, we would need to subtract the *nn'*-junction resistance from the extracted device resistance at high density (see Supplementary Fig. 9). Although we do not have a direct measurement of *nn'*-junction resistance, we suspect that this mechanism provides an explanation of the above mismatch between expected and derived channel resistance.

Evidence of the chiral nature and hence direct evidence of a valley-polarized state in the odd-field configuration are provided by a second split gate pair acting as a valley analyzer. With both gate pairs biased in the same polarization, we expect a conducting state. In case, the second gate pair is biased in opposite polarity, then the resistance should be increased. In this sense, the second gate pair acts as an analyzer of valley polarization of first gate pair. Figure 5 displays the measured resistance for two different

arrangements: a high (low) resistance state appears when two gate pairs have opposite (same) valley polarization. Despite moderate contrast between two gate configurations, these measurements demonstrate the feasibility of gate-controlled valley manipulation at zero magnetic field. Additional experimental details, including alternative gating geometries, are discussed in Supplementary Information.

In conclusion, the improved device performance justifies careful geometry optimization and overall device quality protection measures. Then, pure electrical control of 1D valley-polarized conduction channels in BLG provides a practical pathway to control electron transport in low dimensions. It also paves way to electrically controllable valley transport and manipulation of pseudo-spin. A detailed characterization of valley polarization by exploiting a subsequent gate pair as valley analyzer is part of future investigation.

## Methods

**Electrostatic simulation.** Commercial software MATLAB (partial differential toolbox) is used in electrostatic modeling of dual-split gate configurations. Poisson equation $\nabla \cdot D = \rho_f$ is solved under given boundary conditions. Electron density along BLG surface is then acquired accordingly under conditions with different parameters.

**Device fabrication.** Split bottom gates are fabricated using few-layer graphene flake (less than three layers) exfoliated on $SiO_2/Si$ substrate. EBL (Jeol EBL 6300FX) and oxygen plasma etching (16 W, 30 s) are used to pattern the bottom gates. PMMA 495 A3 (1000 $\mu C\,cm^{-2}$ dose, with proximity effect correction in EBL process, developed in MIBK: IPA (volume ratio 1 : 3) mixture for 30 s) is used as resist for patterning. Few examples of bottom gates are shown in Supplementary Information. The graphitic bottom gates are subsequently cleaned mechanically with Bruker SNL-10 series probe in atomic force microscopy contact mode to remove resist residues. Van der Waals stacking method is exploited: hBN, BLG, and hBN flakes with proper thickness are picked up and release atop bottom gates. Top gates are fabricated in high-resolution mode in Jeol EBL, with alignment mismatch <5 nm (see Supplementary Information). SEM images are captured to check split gap size. Then the device is etched in reactive ion etcher into 1 μm width ribbon. Finally, edge contact technique is used to contact the fully encapsulated device. Field-effect electron mobility is extracted from fitting the curves in Supplementary Fig. 8.

## Data availability

The data that support the findings within this study are available from the corresponding author upon reasonable request.

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

## Acknowledgements
This work was supported by National Research Foundation, Prime Minister's Office, Singapore, under its (R-723-000-018-112), NRF Investigatorship (NRFI Award Number NRF-NRFI2018-08), and Medium-Sized Centre Programme. We thank Fanrong Lin for advice in device fabrication and Alexander Mayorov for valuable discussion.

## Author contributions
J.M. initiated and coordinated the work. H.C. conducted the electrostatic modeling, device fabrication, as well as electrical measurement and data analysis. P.Z. helped with device fabrication. J.L. helped in electrical measurement. J.Q. and B.O. assisted in data analysis. H.C. and J.M. wrote the paper. All authors contributed to the discussions.

## Competing interests
The authors declare no competing interests.
