## [Peer Review File · Nature Communications]

Reviewers' comments:

Reviewer #1 (Remarks to the Author):

In the manuscript, the authors experimentally investigated a purely gate-controlled valley-polarized 1D-channels in bilayer graphene. Surprisingly, the authors highly improved the performance of gated bilayer device by employing optimized geometric dimensions and sample stacking schemes, where the imperfections from device asymmetries and asymmetric gating are notably decreased. Thus, even without the application of external magnetic field, the channel resistance reaches $5.64 \text{ k}\Omega\text{m}$, which is quite close to the expected quantized value ($h/4e^2 \cong 6.45 \text{ k}\Omega\text{m}$).

The device fabrication is impressive and this work deserves to be published in Nature Communication. However, there is a misleading point that should be corrected by the authors. The effect of a perpendicular magnetic field in gated bilayer graphene in the Nature Nanotechnology paper is to effectively suppress the backscattering process due to the imperfection from various sources. Quantum Hall effect has not been involved in the whole process. A systematic theoretical understanding can be found in Frontiers of Physics 14, 23501 (2019), which needs to be properly referenced in this paper by the authors. Additional comments are provided in below.

1. One of the coauthors, Jiabin Qiao, has been acknowledged, which is unreasonable.
2. In fact, some references are not related to this work. But some very related progresses including both experimental and theoretical works are missing, e.g. (a) Nature Communications 7, 11760, 2016; (b) Nano Letters, 11(8), 3453 (2011); (c) Phys. Rev. Lett. 112, 206601 (2014)

After the authors have properly addressed all the mentioned comments, the referee would like to recommend its acceptance in Nature Communications.

Reviewer #2 (Remarks to the Author):

The manuscript reports experiments with valley-polarized electrons in bilayer graphene. It has been theoretically predicted more than ten years ago that if voltages of the opposite signs are applied to the graphene layers, at the edge between these regions topological states appear. These topological states have been previously experimentally observed by other groups. However, at low magnetic field the transport was not ballistic, and the observations were performed in high magnetic fields, close or in to the quantum Hall effect regime. The current manuscript presents improvements which enabled observation of topological channels without a need to go to high magnetic fields.

The information on the journal reads: Nature Communications is an open access journal that publishes high-quality research from all areas of the natural sciences. Papers published by the journal represent important advances of significance to specialists within each field. The manuscript presents high-quality research. Furthermore, topological matter is at the focus of attention of condensed matter physics. Therefore the manuscript is, in principle, eligible for publication in Nature Communications.

However, I do not find that it is currently well-written. Currently, it reads as an incremental extension of the previous works. Instead, the authors should build up an independent story (obviously still citing relevant articles). The argument of the authors is that since they see much higher current if the voltages at the two splits have opposite signs than if they have the same sign, then they must be observing the topological states. The observation is consistent with 4 states carrying one conductance quantum each. Both observations are consistent with the theory. I understand that it is difficult to argue from the experimental data that the states are of topological nature. However, are these conclusions supported by more observations? For example, is Fig. 3b understood in terms of the voltage scales? Is Fig. 3c understood? Why is Fig. 4 not the mirror

image of Fig. 3? The authors need to improve their argumentation to build up the credible story. Significant improvements are needed before publication can be recommended.

Reviewer #3 (Remarks to the Author):

Chen et al. reported electron transport measurement of electric field induced domain wall in AB-stacked bilayer graphene. Quantized conductance was observed for the odd geometry of adjacent split gate regions. Furthermore, signatures of valley-valve were observed in the quadrupole-like gate configurations.

Controlling the valley degree of freedom is of significant interest for valleytronics. Quantum valley Hall effect has been observed in both AB-BA domain walls and electrical field induced domain walls in bilayer graphene. Previous experiments on the latter one were elusive as no quantized conductance was observed at zero magnetic field. A rather high magnetic field has to be applied to make the 1D transport quantized. However, the big magnetic field induces concerns about the interpretation of the valley Hall physics, due to the rendering of regular edge states of standard quantum Hall effect. The current manuscript represents a big improvement in the physics of electrical field induced domain wall as no magnetic field is needed at all! The authors carefully examined possible origins of charge inhomogeneities and successfully addressed some of them. And as a result, much more convincing experimental data was obtained than those observed in previous device structures. What's even more exciting is that the quantized conductance of domain wall corresponds to a mean free path of ~ 700 nm. I would be very interested in learning the true mean-free-path if the channel length is not a limitation. However, I don't think the authors need to provide such data in this manuscript. A lot more efforts should be devoted to this direction, in my opinion.

In sum, I would recommend the publication of this manuscript at Nature Comm.

Reviewer #1 (Remarks to the Author):

In the manuscript, the authors experimentally investigated a purely gate-controlled valley-polarized 1D-channels in bilayer graphene. Surprisingly, the authors highly improved the performance of gated bilayer device by employing optimized geometric dimensions and sample stacking schemes, where the imperfections from device asymmetries and asymmetric gating are notable decreased. Thus, even without the application of external magnetic field, the channel resistance reaches 5.64 k Ω , which is quite close to the expected quantized value ($h/4e^2 \cong 6.45 \text{ k}\Omega \text{hm}$).

The device fabrication is impressive and this work deserves to be published in Nature Communication.

We thank the reviewer for such encouraging and positive feedback.

However, there is a misleading point that should be corrected by the authors. The effect of a perpendicular magnetic field in gated bilayer graphene in the Nature Nanotechnology paper is to effectively suppress the backscattering process due to the imperfection from various sources. Quantum Hall effect has not been involved in the whole process. A systematic theoretical understanding can be found in Frontiers of Physics 14, 23501 (2019), which needs be properly referenced in this paper by the authors.

We thank the author for highlighting the specific reference which has now been included. The Frontier of Physics paper has been properly referenced (please see reply to additional comments).

We also agree that the reference to QHE has been somewhat confusing. The sentences have been modified accordingly:

Abstract: 'Yet, with increasing magnetic field the ungated regions of bilayer graphene will transit into the quantum Hall regime limiting the applications of valley polarized electrons.'

Main text: ‘Yet, the application of high magnetic fields may limit the use of valley-polarized electrons for further studies on electronic transport in the presence of non-zero Berry curvature.’

Additional comments are provided in below.

1. One of the coauthors, Jiabin Qiao, has been acknowledged, which is unreasonable.

Thanks for highlighting this. Indeed, there has been a mistake. This has been modified accordingly.

2. In fact, some references are not related to this work. But some very related progresses including both experimental and theoretical works are missing, e.g. (a) Nature Communications 7, 11760, 2016; (b) Nano Letters, 11(8), 3453 (2011); (c) Phys. Rev. Lett. 112, 206601 (2014)

Thanks for highlighting the references. Indeed, the reviewer mentioned important references which we have included now in the manuscript.

Specifically, in the 2nd paragraph of Introduction, the relevant sentence has been replaced by (The Nano Letter and PRL paper):

‘Theory predicted that^{10,16-18} sign reversal in adjacent regions in bilayer graphene offers a scheme to produce topological valley-polarized 1D-channels.’

In the 3rd paragraph of Introduction part, the reference to is added (Nature Communication paper):

‘Previous work at naturally occurring stacking boundaries demonstrated good contrast between the different electric field configurations^{19,20}.’

Further, we have excluded some previous references in the context of localization in the QH-regime, namely:

1. Probing charging and localization in the quantum Hall regime by graphene p - n - p junctions. *Physical Review B* **81**, 121407
2. Fluctuations and Evidence for Charging in the Quantum Hall Effect. *Physical Review Letters* **82**, 4695-4698
3. The nature of localization in graphene under quantum Hall conditions. *Nature Physics* **5**, 669-674

After the authors have properly addressed all the mentioned comments, the referee would like to recommend its acceptance in Nature Communications. We thank the reviewer for his support. We believe that we have addressed all issues and thank the reviewer for his support.

Reviewer #2 (Remarks to the Author):

The manuscript reports experiments with valley-polarized electrons in bilayer graphene. It has been theoretically predicted more than ten years ago that if voltages of the opposite signs are applied to the graphene layers, at the edge between these regions topological states appear. These topological states have been previously experimentally observed by other groups. However, at low magnetic field the transport was not ballistic, and the observations were performed in high magnetic fields, close or in to the quantum Hall effect regime. The current manuscript presents improvements which enabled observation of topological channels without a need to go to high magnetic fields.

The information on the journal reads: Nature Communications is an open access journal that publishes high-quality research from all areas of the natural sciences. Papers published by the journal represent important advances of significance to specialists within each field. The manuscript presents high-quality research. Furthermore, topological matter is at the focus of attention of condensed matter physics. Therefore, the manuscript is, in principle, eligible for publication in Nature Communications.

We thank the reviewer for this positive assessment of our work.

However, I do not find that it is currently well-written. Currently, it reads as an incremental extension of the previous works. Instead, the authors should build up an independent story (obviously still citing relevant articles).

As the reviewer states, our experiments constitute a major improvement to published experimental data.

The qualitatively new result is demonstration of chiral states at \$B = 0T\$. As stated in the manuscript, this will allow experiments without having the bilayer leads in the QHE and dominated by edge state transport.

We explain this as a result of improved device fabrication schemes together with a quantitative estimation of the effects of sample imperfections.

In order to further strengthen the paper, we have included now additional measurements with the device in valley analyzer configuration (see below).

The argument of the authors is that since they see much higher current if the voltages at the two splits have opposite signs than if they have the same sign, then they must be observing the topological states. The observation is consistent with 4 states carrying one conductance quantum each. Both observations are consistent with the theory. I understand that it is difficult to argue from the experimental data that the states are of topological nature.

However, are these conclusions supported by more observations?

Real proof for topological states would be the demonstration of a valley analyzer.

The experimental challenge hereby is that two high quality junctions are required. In order to further strengthen the paper, we decided to show additional data demonstrating the chiral character of the states in fig 5 (and data in SI). Please see Figure 5 in main text.

The device in analyzer regime consists of two gate-pairs (each comprises of four gates) which are individually configured to generate chiral states in the same or opposite valley polarization.

In practice, we fix all gate voltages on the 1st gate-pair (the left one in device schematics) to remain in charge neutral valley polarized state (e.g., point b in Supplementary 12) throughout measurements. The potentials on 2nd gate-pair is varied with the same method shown in Fig. 3&4, that is, half gate-pair is kept at charge neutral point, the other half is applied with varying voltages on top gate when bottom gate is at different fixed potentials. The data of Fig. 5 are thus acquired from two measurements: one of them is measured with half gate-pair at positive D -field, the other one at negative D -field.

Please note that the reason why we do not combine two diagonal gates in 2nd pair to measure the above two state in one measurement is that charge neutrality points in upper and lower parts are not always the same. This may also be the cause why we do not achieve so highly resistances when the 2nd gate pair is in even-configuration (insulating state). Please see Supplementary Information part 8 for more details.

Fig. 5 Demonstration of chiral nature: Valley analyzer measurement with two gate-pairs in same (left) or opposite (right) valley-polarized state. Gate configurations are shown underneath. During the measurement, the 1st gate-pair is kept at constant voltages to maintain in a fixed valley polarized state. One half of the 2nd gate pair is at fixed, too and only one top-gate and one bottom gate is scanned (similar scheme to fig 3,4). The BLG resistance as function displays two different regimes: High (low) resistance states are observed under opposite (same) valley polarized state of two gate-pairs, indicating the chiral nature of states in odd-gate configuration. The inset above measurement curves shows exemplary 3D side-view of gate configurations.

For example, is Fig. 3b understood in terms of the voltage scales?

We thank the reviewer for highlighting this point which was not explained in the manuscript. We estimate induced charge carrier density and displacement field from the capacitive coupling as follows:

The relations can be derived:

$$n = \varepsilon_0 \varepsilon_r ((V_t - V_{t0})/d_t + (V_b - V_{b0}/d_b))/e,$$

$$D = \varepsilon_r ((V_t - V_{t0})/d_t - (V_b - V_{b0}/d_b))/2,$$

where ε_0 , ε_r , V_t , V_{t0} , V_b , V_{b0} , d_t , d_b , e , denotes vacuum electric constant, dielectric constant, top-gate voltage, top-gate voltage offset, bottom gate voltage, bottom gate voltage offset, top/bottom dielectric thickness and elementary charge. The dielectric constant value of hBN used in the device is 4 from following reference:

‘Boron nitride substrates for high-quality graphene electronics. *Nature Nanotech* **5**, 722–726 (2010).’

The dielectric thickness for top and bottom hBN layer is measured from atomic force microscopy. And offsets for top/bottom gates here originate from disorder induced doping, either from lithography or heterostructure inhomogeneity. The specific details can be found in SI part 4.

Is Fig. 3c understood?

Fig 3c highlights the odd-configuration. We expect the formation of chiral states resulting in a saturation of resistance. Importantly, we show here the raw data without any correction to series resistance caused by contact resistance and resistance of bilayer graphene.

As described in the main text, we can estimate the series resistance as the measured value at high density. Subtracting this value from the raw data yields a number very close to the expected quantum resistance.

Why is Fig. 4 not the mirror image of Fig. 3?

The reviewer hints on an important point, which we hope to make clearer in the updated version of the manuscript.

Since the gate-pair corresponds to a quadrupole-like arrangement, there are two mirror symmetries: top-down mirror image and left-right mirror image. In fig 4, we chose to compare the top-down mirror with respect to the gate configuration in fig 3. For completeness, we include now left-right mirror configurations in SI.

The authors need to improve their argumentation to build up the credible story. Significant improvements are needed before publication can be recommended.

We added two sets of important data which highlight the performance of our devices and demonstrate the chiral character of 1D-states:

- In SI we include now additional ‘mirror-image’ results as proof of gate controllable states.
- Moreover, direct evidence for chiral states are added by measurement data with the device in valley-analyzer configuration.

We are confident that the additional data strengthens the paper significantly, and sincerely hope that the reviewer finds this convincing as well.

Reviewer #3 (Remarks to the Author):

Chen et al. reported electron transport measurement of electric field induced domain wall in AB-stacked bilayer graphene. Quantized conductance was observed for the odd geometry of adjacent split gate regions. Furthermore, signatures of valley-valve were observed in the quadrupole-like gate configurations.

Controlling the valley degree of freedom is of significant interest for valleytronics. Quantum valley Hall effect has been observed in both AB-BA domain walls and electrical field induced domain walls in bilayer graphene. Previous experiments on the latter one were elusive as no quantized conductance was observed at zero magnetic field. A rather high magnetic field has to be applied to make the 1D transport quantized. However, the big magnetic field induces concerns about the interpretation of the valley Hall physics, due to the rendering of regular edge states of standard quantum Hall effect. The current manuscript represents a big improvement in the physics of electrical field induced domain wall as no magnetic field is needed at all! The authors carefully examined possible origins of charge inhomogeneities and successfully addressed some of them. And as a result, much more convincing experimental data was obtained than those observed in previous device structures.

We sincerely thank the reviewer for this very positive feedback.

What's even more exciting is that the quantized conductance of domain wall corresponds to a mean free path of ~ 700 nm. I would be very interested in learning the true mean-free-path if the channel length is not a limitation.

We should recall that the gate length in our device is ~ 400 nm. We have estimated the mean free path from analysis of standard transport measurements, i.e. without operating the local gates.

We agree that it would be interesting to measure a series of devices with increasing gate length to observe a crossover from ballistic transport to backscattered dominated devices. Yet, this would imply a tremendous amount of additional work which we hope to avoid.

However, I don't think the authors need to provide such data in this manuscript. A lot more efforts should be devoted to this direction, in my opinion.

We agree with the reviewer that further characterization of device parameters and performance is required to enable further experiments and to strive towards applications. Yet, this is beyond the scope of this paper.

In sum, I would recommend the publication of this manuscript at Nature Comm. We thank the reviewer for this very positive statement.

Manuscript change list:

1. At page 2, in Abstract part, original sentence ‘It has been predicted that signal reversal of...’ is modified to ‘Signal reversal of...’.

Also, the reference to QHE is revised as:

‘Yet, with increasing magnetic field there is a transition to the quantum Hall regime in the bilayer leads which could limit the applications of valley polarized electrons.’

Moreover, reference to conductance of valley polarized state is revised as:

‘The valley polarized state displays conductance of nearly $4e^2/h$ and produces contrast in a subsequent valley analyzer configuration.’

2. At page 3, in 2nd paragraph in Introduction part. Additional theoretical works on valley channels are referred and the sentence is replaced by:

‘Theory predicted that^{10,16-18} sign reversal in adjacent regions in bilayer graphene (BLG) offers a scheme to produce topological valley-polarized 1D-channels.’

In this paragraph some expressions are rephrased and marked with red in manuscript.

3. At page 4, in 3rd paragraph in Introduction part. Additional work on topological edge states is referenced:

‘Previous work at naturally occurring stacking boundaries demonstrated good contrast between the different electric field configurations^{19,20}.’

4. At page 4, the previous reference to QHE has been removed and the sentence is replaced by:

‘Yet, the application of high magnetic fields may limit the use of valley-polarized electrons for further studies on electronic transport in the presence of non-zero Berry curvature.’

Moreover, at the end of the paragraph, we add:

‘Evidence of chiral nature of the 1D-channels is provided by measurements in valley analyzer configuration.’

5. At page 5, an additionally work on lateral confinement induced sub-band is referenced (ref. 22). Some expressions have been rephrased and marked with red color.

6. At page 6, a sentence has been replaced by:

‘BLG is also assumed to be a metal which will yield an upper bound of the induced charge.’

Some sentences have been rephrased and marked with red color and highlighted in yellow.

7. At page 7, the fabrication process has been replaced by:

‘The actual device is fabricated by stacking hBN, BLG, hBN layer subsequently and release this stack atop pre-patterned few-layer graphene split gates. Samples are designed to have 4 gate-pairs to increase yield, and to allow for valley analyzer experiments. A dry van der Waals stacking method is exploited to reduce inhomogeneity and disorder in the heterostructure. Generally, the split channel width of around 50nm is fabricated. The graphene bottom gates are etched in O₂-plasma using a PMMA mask. We have optimized the fabrication to accommodate dimensional changes in the real device with respect to the design. For example, etching of the graphene bottom gates affect the width of the gates depending on the etch time, concentration etc. Further, we exploit slight overdose for the top gates to minimize PMMA residues but increases the gates width of 15-20 nm. Hence to achieve a nominal gates spacing of 50 nm, we use a 35 nm gap for bottom layer and 65 nm gap for top layer. Edge contacts²⁶ is used to contact the fully encapsulated BLG. The design of the top gates is adjusted as discussed above. Further details of device fabrication are included in Methods and Supplementary Information.’

8. At page 8, the 1st paragraph of results part. Additional data is provided for device 1 as follows:

‘The mean free path of electron is estimated to be around 330 nm and 700 nm for device 1 and 2, respectively (see SI).’

Some sentences have been rephrased and marked with red color.

9. At page 9, the relations of top/bottom gate voltages to charge carrier density and displacement field is added as:

‘Here the relations of n and D -field can be calculated from $n = \epsilon_0 \epsilon_r ((V_t - V_{t0})/d_t + (V_b - V_{b0}/d_b))/e$, $D = \epsilon_r ((V_t - V_{t0})/d_t - (V_b - V_{b0}/d_b))/2$. Here, ϵ_0 is vacuum electric constant, ϵ_r dielectric constant of hBN, $V_{t(b)}$ top(bottom)-gate voltage, $V_{t0(b0)}$ top(bottom)-gate voltage offset, $d_{t(b)}$ top (bottom) thickness of dielectric, and e the elementary charge. (Further details are provided in SI)’

Some sentences have been rephrased and marked with red color.

10. At page 10, the context on Fig. 4 and additional devices is moved forward from discussion part to here.

11. At page 10, the subtracted channel resistance is slightly modified from 5.64 kOhm to 5.72 kOhm after slight correction.

12. From page 10, in the first and second paragraph in Discussion part, some changes have been made and marked with red color and highlighted in yellow.

13. At page 11, discussion on p-n junction effect on the BLG resistance have been modified. Three previous references are deleted and replaced with new ones.

References deleted:

1. Probing charging and localization in the quantum Hall regime by graphene p - n - p junctions. *Physical Review B* **81**, 121407

2. Fluctuations and Evidence for Charging in the Quantum Hall Effect.
Physical Review Letters **82**, 4695-4698
3. The nature of localization in graphene under quantum Hall conditions.
Nature Physics **5**, 669-674

Reference added:

1. Quantum Hall Effect in a Gate-Controlled p-n Junction of Graphene.
Science **317**, 638

Moreover, citation of ref. 30 and 31 has been moved to end of the paragraph. The modified paragraph now is:

‘This has been observed experimentally³² in pn-junctions displaying higher resistance in the pn-state, compared to nn’-state. In locally gated bilayer channels, the 2-terminal resistance depends on the combination of p- and n-regions and locally induced gaps^{30,31}.’

14. At page 12, discussion on valley analyser measurement in Fig. 5 is added:

‘Evidence of the chiral nature, and hence direct evidence of a valley polarized state in the odd-field configuration is provided by a second split gate pair acting as valley analyzer. With both gate pairs biased in the same polarization, then we expected a conducting state. In case, the second gate pair is biased in opposite polarity, then the resistance should be increased. In this sense, the second gate-pair acts as analyzer of valley polarization of first gate-pair.’

Fig. 5 displays the measured resistance for two different arrangements: a high (low) resistance state appears when two gate-pairs have opposite (same) valley polarization. Despite moderate contrast between two gate configurations, these measurements demonstrate the feasibility of gate-controlled valley manipulation at zero magnetic field.’

15. In Acknowledgement part, a coauthor’s name is deleted.

16. In Author contribution part, several corrections are made, and it is now:
'J.M. initiated and coordinated the work. H.C. conducted the electrostatic modelling, device fabrication, as well as electrical measurement and data analysis. P.Z. helped with device fabrication. J.L. helped in electrical measurement. H.C. and J.M. wrote the paper. All authors contributed to the discussions. '

17. Captions of Fig.1, Fig. 2, Fig. 3 and Fig. 4 are modified. The schematics in Fig. 1b is modified. The x-axis and y-axis of Fig. 1c and the x-axis of Fig. 1d has been updated with correct scale.

18. Fig. 5 is added, as a proof for the topological nature of the valley polarized state.

REVIEWERS' COMMENTS:

Reviewer #2 (Remarks to the Author):

I am satisfied with the responses of the authors to the referee reports (including mine) and the introduced changes, and I can now recommend the manuscript for publication.